# Exploring Chinese Elderly’s Trust in the Healthcare System: Empirical Evidence from a Population-Based Survey in China

**DOI:** 10.3390/ijerph192416461

**Published:** 2022-12-08

**Authors:** Lu Chen, Miaoting Cheng

**Affiliations:** 1School of Journalism and Communication, Guangzhou University, Guangzhou 510006, China; 2Department of Educational Technology, Faculty of Education, Shenzhen University, Shenzhen 518060, China

**Keywords:** elderly, trust, healthcare system, China Social Survey

## Abstract

This research aims to investigate how much the Chinese elderly trust the healthcare system and the critical factors that influence their trust. We use data from the China Social Survey (CSS) collected by the Chinese Academy of Social Sciences in the year 2019 to examine how demographic factors, social-economic status, internet access, and perceptions of the healthcare system impact the Chinese elderly’s trust in the healthcare system. Our research finds male gender, high educational level, and having internet access are negatively related to the elderly’s trust in the healthcare system. Our research also reveals that the elderly’s trust in the healthcare system was significantly related to their subjective perception of their social–economic status, upward mobility, and perception of accessibility and affordability rather than other objective indicators such as income and financial protection. The results imply that the elderly have a pessimistic expectation of their subjective social status and future possibilities of upward mobility in their later life, which deepens their distrust of the health system. Additionally, the accessibility and affordability of the healthcare system have remained problematic among the Chinese elderly. The study provides important theoretical and practical implications to enhance the elderly’s trust in the healthcare system.

## 1. Introduction

The healthcare system in China has undergone radical changes in the post-reform era. China’s healthcare system went through the first phase of market-oriented reform from 1986 and the second shift toward reemphasizing the government’s role after 2002 [1]. In the 1980s, the government reduced financial support to the healthcare sector, and public hospitals had to be financially self-sufficient and profit-oriented, which further caused various problems in access, service quality, and regulation [2]. Although the social health insurance coverage and capacity of public health services increased from 2003 to 2011, the quality of healthcare services has not improved greatly, the cost of medical care is still increasing, and the health resources are unevenly distributed [3].

As a response to public discontent, in 2009, China launched healthcare system reform and has achieved substantial improvements in access to healthcare services and financial protection. However, the public’s satisfaction with the healthcare system did not increase from 2010 to 2016 [4]. The challenges remain in the quality of care, control of non-communicable diseases, burden of the aging population, efficiency in delivery, control of healthcare expenditures, and public satisfaction [5]. Many studies have started to focus on trust, which sometimes is treated as a part of public perception or a dimension of satisfaction with the healthcare system and healthcare service providers and physicians [6,7]. However, among the existing empirical studies, few examine the Chinese elderly’s trust in the healthcare system. In our research, we aim to investigate the following research questions: (1) How much do the Chinese elderly trust the healthcare system? (2) What are the critical factors that influence Chinese elderly’s trust in the healthcare system? We use data from the China Social Survey (CSS) collected by the Chinese Academy of Social Sciences in the year 2019 to examine how demographic factors, social–economic status, internet access, and perceptions of the healthcare system have an impact on Chinese elderly’s trust in the healthcare system.

## 2. Literature Review

### 2.1. Definition and Conceptualization of Trust

Although the definitions of trust vary according to different disciplines, such as sociology, psychology, and risk management, they share something in common. First, it is related to vulnerability. The majority of studies stress the optimistic acceptance of a vulnerable situation in which the truster believes the trustee will care for the truster’s interests [8]. The trust literature distinguishes trustworthiness (the ability, benevolence, and integrity of a trustee) and trust propensity (a dispositional willingness to rely on others) from trust (the intention to show vulnerability to a trustee based on positive expectations of his or her actions) [9]. Second, it is related to social relationships. Trust is a social relationship in which principals—for whatever reason or state of mind—invest resources, authority, or responsibility in another to act on their behalf for some uncertain future return [10]. Third, trust is related to risk. Trust is a substitute for knowledge and adaptive response to uncertain futures and incalculable risks [11]. If knowledge is missing, trust is used to assess the benefits and the risks associated with a hazard [12]. Trust can be a solution to risk. Risk and trust can be two contrasting and sometimes overlapping ways of coping with vulnerabilities amidst uncertainty. Within an organization, tension exists between trust and risk and, sometimes, the risk can be managed through trust [13]. Trust is multi-leveled from personal trust in individuals to generalized trust in the whole social order [14].

Trust in healthcare also has different levels; sometimes, these levels are entangled. In healthcare, people are placed as patients and express themselves as members of the public [15]. Healthcare systems are an important means by which individuals interact with their government. Thus, healthcare system performances, such as quality of healthcare services, service delivery, fair treatment, better health outcomes, and financial risk protection, are associated with public trust in government [16]. Trust has a changing relationship. When personal judgement of risk increases, trust in experts, institutions, healthcare organizations, the medical professions in general, and healthcare systems declines [17]. Generally speaking, interpersonal trust is considered essential for effective doctor–patient communication, and institutional trust or public trust in healthcare systems is important for public support.

### 2.2. Trust Crisis in the Healthcare System in China

The healthcare system in China has suffered from a trust crisis among the public since the 1990s. The distrust in primary care causes people to seek secondary and tertiary care even for minor, self-limiting conditions, and distrust in medical professionals causes violence against doctors [18]. Physicians’ distrust of patients and their relatives leads to increased levels of fear and self-protection by doctors, which exacerbate communication difficulties; in turn, this increases physician workloads, adding to a strong sense of injustice and victimization [19]. Public hospitals make up the largest slice of the healthcare sector and they are regulated by the state. In addition, because the reform in the healthcare system was initiated by the state, people’s trust in the healthcare system then reflects the public trust in government and the state to some extent and attitudes towards social equity and justice issues. Though public trust in local government is higher than in many other countries, Chinese people have low public trust in the healthcare system, which is positively related to trust in other people, self-reported happiness, self-reported health status, and attitude towards social equity [20].

Current studies on trust in healthcare adopt two perspectives. The macro one usually involves national or international surveys to examine the social and demographic factors that impact on public trust in the healthcare system, which is related to people’s assessment of the quality, accessibility, and efficiency of medical services. The empirical studies find that public assessment of trust tends to address the views of healthcare at the micro level about the quality of healthcare rather than broader concerns of citizens with how the services are run and paid for [21]. Data from the International Social Survey Program in 2011 and 2013 in 31 countries exhibit that one of the strongest predictors of trust was the respondents’ most recent healthcare experience [22]. Previous empirical studies suggested that demographic factors such as gender, age, and educational level were significantly related to Chinese people’s trust in the healthcare system [23].

The micro perspective focuses on organizational or institutional trust in particular hospitals or interpersonal trust in particular doctors. Organizational trust and interpersonal trust are complements rather than substitutes [24]. Patients’ trust in their physician is related to having a choice of physicians, having a longer relationship with their physician, and trusting their managed care organization [25]. As China is currently promoting primary care in community healthcare centers, many studies discuss how patients’ trust could influence their choice of community healthcare services or hospitals. The patient trust crisis has continued to inhibit the development of community-based primary care [26]. A study finds that the ownership of community healthcare centers significantly influences patients’ trust in community health services [27]. Doctor–patient trust will stimulate patients to choose primary healthcare institutions [28].

China has been experiencing a widespread and profound crisis of patient–physician trust [29]. Thus, many studies on China’s healthcare tend to explore the patient–doctor relationship and communication. In China’s healthcare sector, there is a popular and socio-culturally distinctive phenomenon known as “guanxi jiuyi” (medical guanxi), whereby patients draw on their “guanxi” (personal connections) with physicians when seeking healthcare [30]. Low overall trust and satisfaction among patients can negatively influence patients’ self-management and doctors’ enthusiasm [31]. Higher trust scores were associated with a lower likelihood of a serious response to medical disputes, including violence against healthcare workers [32].

As the proliferation of the internet has dramatically changed Chinese society, the influence of exposure to online information on trust in the healthcare system has become an area worth studying. Patient distrust in physicians is the information asymmetry between patients and physicians, which is a natural property of the physician–patient relationship [33]. The internet provides people with more information on public issues and exposure to negative information. Some studies started to explore the influence of media and online information on patients’ trust. The media in China tends to report adverse news on unaffordable healthcare and inappropriate prices of medicines, which have negatively affected the perceptions of the doctor–patient relationship and led to public mistrust in doctors [34]. Public trust in medical professions has declined in recent decades and is related to health information acquisition from various sources, including new media [35]. The lack of credible mechanisms to verify medical information online has made internet users vulnerable to deception. The internet fails to provide trustworthy medical knowledge and alleviate the heightened tension between patients and doctors [36]. When the online information for self-diagnosis contradicts the physicians’ diagnosis, the trust of patients in their physicians will reduce [37]. Trust also impacts patients’ selection of physicians in online healthcare communities [38]. If patients show a high level of trust when using online healthcare services, they are more likely to continue their usage behavior [39].

### 2.3. Chinese Elderly and the Healthcare System

According to the Seventh National Population Census, China is experiencing a rapidly aging population [40]. Older people are a vulnerable group confronting a higher risk of chronic diseases and declination of health conditions. The health risk of the elderly will cause rising demand for medical services and medical expenditure from 2020–2060 [41]. With the demographic transition, the aging population poses greater challenges to China’s public health than ever. The absolute number of the elderly population in China is very large, the growth in the aging population is rapid, and the growth in the population above 80 is apparent [42]. China has the largest elderly population in the world and is one of the most rapidly aging societies, which calls for the provision of affordable and accessible healthcare services [43].

In China, the discontent from the public towards accessibility and affordability of healthcare services is commonly known as “Kanbingnan, Kanbinggui” in Chinese, which literally means “It is difficult to get access to medical services and the cost of medical services is unaffordable” [44]. As existing empirical studies showed, “Kanbingnan, Kanbinggui” gained serious political attention at the end of the 1990s because of widespread social protests of health problems and a push for the government to launch healthcare reform [45]. “Kanbingnan, Kanbinggui” has frequently appeared in government policy documents referring to the accessibility and affordability of healthcare services. The Chinese elderly are desperate for accessible and affordable healthcare services. Among China’s huge elderly population, a relatively high proportion of the elderly have chronic diseases and a considerable portion of the elderly are in poor health condition, which creates a huge demand for healthcare [46]. The elderly with poor health status will increase the healthcare utilization and healthcare expenditure of the whole family [47]. A comparative study on healthcare utilization and affordability in Zhejiang and Gansu Provinces among the elderly has found that the older people in both provinces still face a heavily out-of-pocket burden and high proportion of pharmaceutical spending [48]. Moreover, the traditional cultural narratives devalue the healthcare demands of the elderly as “unworthy of care and treatment”, which were internalized and downgrade their expectation of receiving medical care [49], which further leaves them in difficult situations.

To recap, the review of the literature showed that a number of studies have set out to explore the public’s trust in the healthcare system, and thus some potential influential factors were proposed. Nevertheless, though elders are among the most vulnerable groups demanding healthcare services in China, limited effort has been made to investigate trust in the healthcare system across elders with different demographics, social economic statuses, access to the internet, and perceptions of the healthcare system.

## 3. Method

### 3.1. Data and Sample

Data were derived from the open database of the Chinese Social Survey (CSS). The CSS is a nationally representative large-scale cross-sectional survey project launched by the Institute of Sociology of the Chinese Academy of Social Sciences. This study used the original dataset of the CSS 2019, which covered 10,283 individuals across 604 villages/communities and 151 counties/districts from 31 provinces/cities/autonomous regions throughout China. As this study considers only Chinese elders, the inclusion criteria for respondents were those aged 60 years old or above (born in or before 1959). We further screened, matched, and processed the variables recording the birth year of the respondents in the questionnaire and finally obtained 2282 sample data.

### 3.2. Variables

#### 3.2.1. Dependent Measure

The dependent variable of this study is trust in the healthcare system. Respondents were requested to provide responses to the question: “Do you trust the following institutions?” These 12 institutions not only cover the official apparatus, public sectors, and private sectors, but also contain specific workplaces of the respondents. Besides the components of the healthcare system such as hospitals, the other institutions include the central government; government at district or county level; government at town level; organizations such as labor unions, the Communist Youth League, and the Women’s Federation; working unit or company; charity organizations; media; bank; insurance company; court; police. In this question, they rated their trust in 12 institutions on a 4-point Likert scale ranging from “complete distrust”, “distrust”, “trust”, to “complete trust”, respectively. One item that measured respondents’ trust in the healthcare system was used as the dependent measure. A binary variable was then constructed using the responses from the item, assigning a value of 1 (“trust”) if the response was “fairly trust” or “completely trust” and a value of 0 (“distrust”) if the response was “completely distrust” or “fairly distrust”. Other items measuring elders’ trusts in other institutions were also constructed as binary variables to compare elders’ trust in different institutions. In order to reflect the attitude of respondents more accurately, we excluded the “not sure” response from analyses because the respondent exhibited neither a firm positive nor negative attitude.

#### 3.2.2. Independent Measures

Four sets of independent variables were included to explore the determinants of respondents’ trust in healthcare. The first set is demographic variables, including age, gender, education level, and total annual income. Respondents’ age was derived as the birth date subtracted from 2019 in the CSS 2019 data. Gender is a binary variable, assigning a value of 1 to male respondents and a value of 0 to female respondents. Respondents’ education level was categorized into four levels, assigning a value of 1 to “primary education or below”, a value of 2 to “junior high school”, a value of 3 to “high school”, and a value of 4 to “university specialist or above”. Respondents’ total annual income was developed from one item: “Please tell us what your total annual income was in 2018” where they were requested to fill in the exact amount of annual income. Age and total annual income were treated as continuous variables, whereas gender and education level were treated as categorical variables.

The second set is self-rated socio-economic status (SES) variables, including upward subjective SES, perceived mobility of SES, and mobility expectation of SES. Subjective SES refers to a person’s self-perceived current status in the social status hierarchy [50]. In this study, we used self-rated current SES to represent subjective SES. One item was used to measure the current SES of respondents based on: “Which level of SES do you think that you currently belong to in your living place?” Perceived mobility of SES refers to the self-rated upward or downward changes in current SES compared to the SES in the past. Elders’ SES mobility was derived as the current SES subtracted from the past SES and the results were categorized into three levels, assigning a value of 1 to “downward mobility”, a value of 2 to “unchanged SES”, and a value of 3 to “upward mobility”. One item was used to measure the past SES of respondents based on: “Five years ago, which level of SES do you think that you belonged to in your living place?” Perceived mobility expectation of SES refers to the expected upward or downward changes in current SES in the future. Elders’ SES mobility expectation was measured using one item: “In the next five years, which level of SES do you think you will belong to in your living place?” Elders self-rated the three items with the responses ranging from “upper”, “middle-upper”, “middle”, “middle-lower”, to “lower”. The responses were categorized into three levels, assigning a value of 1 to “lower” if the response was “middle-lower” or “lower”, a value of 2 to “middle”, and a value of 3 to “upper” if the response was “upper” or “middle-upper”. SES variables were treated as continuous variables.

The third set is internet use variables, including internet access and trust in the internet. The CSS survey used one item to measure respondents’ internet access based on: “With the proliferation of the internet, will you get online (e.g., using computer or mobile phone to read news, use WeChat) in your daily time?” The CSS survey constructed internet access as a binary variable, assigning a value of 1 if the response was “Yes” and a value of 0 if the response was “No”. In terms of trust in the internet, respondents were requested to respond to four items concerning their trust in the internet with responses on a 4-point Likert scale ranging from 1 (“strongly agree”), 2 (“fairly agree”), 3 (“fairly disagree”), to 4 (“strongly disagree”). Sample items included “Internet is the most trustworthy channel to reflect citizens’ opinions and the true situation of the society”. We reversed the values of responses to keep them consistent with the meaning ranging from 1 (“strongly disagree”) to 4 (“strongly agree”), and developed an understanding of elders’ trust in the internet using the mean value of the four items. Internet access was treated as a categorical variable, whereas trust in the internet was treated as a continuous variable.

The fourth set is perceptions of the healthcare system and financial protection variables, including perceived fairness in healthcare, insurance status, and perceptions of healthcare services. The CSS used one item to measure respondents’ perceived fairness in the healthcare system: “What do you think about the fairness of the following aspects of social life: public healthcare system?” with responses on a 4-point Likert scale ranging from 1 (“very unfair”), 2 (“fairly unfair”), 3 (“fairly fair”), to 4 (“very fair”). We only included respondents’ old-age and medical insurance to measure elders’ insurance status since these two insurances are closely related to their participation in the healthcare system. One item was used to ask respondents whether they own old-age insurance or medical insurance, respectively. Two binary variables were then constructed, assigning a value of 1 if the respondents owned the insurance and a value of 0 if not. The CSS survey requested respondents to select the most pressing social problems in China, and “Kanbingnan, Kanbinggui” was listed as one of the problems. We used this item to measure respondents’ perceived accessibility and affordability of the healthcare system. A binary variable was then constructed, assigning a value of 1 if this problem was selected and a value of 0 if not. Perceived fairness in healthcare was treated as a continuous variable, whereas the other variables were treated as categorical variables.

#### 3.2.3. Statistical Analysis

First, descriptive analysis was used to explore the holistic profile of demographics, SES, internet use, and perception of the healthcare system of elders. Then, the chi-square test for independence or independent-sample *t*-test analysis was employed to examine how elders’ trust in the healthcare system was significantly different in terms of the four sets of variables given the attributes of the variables. Finally, since the dependent variable, trust in the healthcare system, is presented as a binary variable, we adopted the binary logistic regression analysis to explore the effects of the four sets of independent variables on elders’ trust in the healthcare system. These data analyses were conducted using IBM SPSS 25.

## 4. Findings

### 4.1. Descriptive Characteristics of the Sample

Table 1 shows the descriptive statistics of the variables of interest in our sample. The final sample consisted of 1197 males (47.5%) and females (52.5%), with an average age of 64.59 (SD = 2.74). Most elderly have an education level of junior high school or below (81.2%), and only 3.1% of them have received a university specialist degree or above. It is also noteworthy that 72.6% of the elderly did not have internet access in their daily life and over half of them perceived the healthcare system as difficult to access and unaffordable (59.0%). We further conducted a descriptive analysis of the elderly’s trust in the 12 different institutions and the results showed that the elderly’s trust in the healthcare system (68.2%) was only ranked slightly higher than that in the insurance company (64.1%), calling for attention to a crisis of the elderly’s trust in the healthcare system.

### 4.2. Disparities in Trust in Healthcare

The chi-square test for independence and independent-sample *t*-test were conducted to explore the association between the four sets of variables and the elderly’s trust in healthcare. Table 2 summarizes and presents the descriptive statistics of key results. The results of the chi-square test showed that the elderly’s trust in the healthcare system was significantly associated with gender (χ^2^ = 33.38 ***, Phi = 0.125), educational level (χ^2^ = 46.97 ***, Phi = 0.148), internet access (χ^2^ = 33.38 ***, Phi = −0.164), and “Kanbingnan, Kanbinggui” (χ^2^ = 31.32 ***, Phi = −0.121), but had no significant association with old-age insurance (χ^2^ = 0.76, Phi = 0.02) and medical insurance (χ^2^ = 0.13, Phi = −0.01). More specifically, the results showed that female elderly (55.7%), the elderly with primary education or below (58.3%), and no internet access (76.7%) were more likely to trust in the healthcare system, whereas the elderly who perceived “Kanbingnan, Kanbinggui” were less likely to trust the healthcare system (31.5%).

The results of the *t*-test showed that the elderly’s trust in the healthcare system was significantly different in all the continuous variables, except for age (t = 1.17, *p* = 0.24) and SES upward mobility (t = 0.61, *p* = 0.54). Given total annual income violated the assumption of normality, it was not included in the *t*-test analysis. More specifically, the elderly who trust the healthcare system were found to perceive higher current SES (M = 1.64, SD = 0.64) and higher SES upward mobility expectation (M = 1.89, SD = 0.78) than their counterparts (M = 1.55, SD = 0.66; M = 1.78, SD = 0.79). The elderly who trust the healthcare system were also found to have higher scores for trust in the internet (M = 2.81, SD = 0.59) than their counterparts (M = 2.68, SD = 0.62). Moreover, the elderly who trust the healthcare system had higher scores for perceived fairness in the healthcare system (M = 3.05, SD = 0.77) than their counterparts (M = 2.29, SD = 0.85). It is also reasonable that the elderly who demonstrated distrust in the healthcare system had higher scores for “Kanbingnan, Kanbinggui” (M = 0.68, SD = 0.46) than those who trust the healthcare system (M = 0.55, SD = 0.50).

### 4.3. Logistic Regression Model of Trust in the Healthcare System

To further examine the factors influencing the elderly’s trust in the healthcare system, the binary logistic regression analysis was used to investigate the independent variables’ effects on the elderly’s trust in the healthcare system, including demographic variables (Model 1), SES variables (Model 2), internet use variables (Model 3), and perceptions and financial protection variables (Model 4). The results of the Hosmer–Lemeshow chi-square test showed that Model 1 (χ^2^(8) = 8.66, *p* = 0.37), Model 2 (χ^2^(8) = 9.86, *p* = 0.28), Model 3 (χ^2^(8) = 3.04, *p* = 0.93), and Model 4 (χ^2^(8) = 5.30, *p* = 0.73) were all insignificant, indicating all the models had a good fit. Moreover, the results of the likelihood ratio chi-square test showed that Model 1 (χ^2^(6) = 57.60, *p* < 0.001), Model 2 (χ^2^(9) = 66.12, *p* < 0.001), Model 3 (χ^2^(11) = 86.35, *p* < 0.001), and Model 4 (χ^2^(15) = 293.86, *p* < 0.001) all have a significant improvement in fit over the intercept-only null model, with the latter models having better explanatory power. The beta (B), standard error (SE), odd ratios (exp(B), OR), and the associated 95% confidence interval (CI) were reported for each regression (see Table 3).

For Model 1, only demographic variables were included. The results showed that age and total annual income did not significantly affect the elderly’s trust in the healthcare system. Education level and gender (B = −0.47, OR = 0.63, *p* < 0.001) were significant and negative predictors of the dependent variable. More specifically, using primary education or below as the reference (B = −0.64, OR = 0.53, *p* < 0.001), for every one unit increase in the elderly’s education level, the odds ratio of increasing the elderly’s trust by one additional level decreased by 43% (i.e., (exp(B) − 1)%). The results indicated that the elderly with primary education or below were more likely to trust the healthcare system than the elderly with high school education. Using female gender as the reference group (B = −0.47, OR = 0.63, *p* < 0.001), for every unit increase in gender, the odds ratio of increasing the elderly’s trust by one additional level decreased by 53%, indicating male elderly were less likely to trust in the healthcare system.

For Model 2, SES variables were further included in the model. In terms of demographic variables, similar findings were found, with gender (B = −0.46, OR = 0.63, *p* < 0.001) and education level (B = −0.63, OR = 0.53, *p* < 0.001) being significant and negative predictors of the elderly’s trust in the healthcare system, whereas age and total annual income had no significant effect on the dependent variable. It is worth noting that SES variables did not significantly affect the elderly’s trust in the healthcare system.

For Model 3, we further included internet use variables in the model. Similar findings were found regarding the effects of age, total annual income, gender, and SES variables on the dependent variable, with age, total annual income, and SES variables still having no significant effect, whereas gender was a significant and negative predictor. However, we found that the association between education level and the dependent variable disappeared when internet use variables were introduced into Model 3. Internet use variables were found to have strong effects on the elderly’s trust. More specifically, internet access was a significant and negative predictor (B = −0.58, OR = 0.56, *p* < 0.001), indicating that every unit increase in elders’ internet access lowered the odds ratio of increasing the elderly’s trust by one additional level by 44%. In contrast, the elderly’s trust in the internet was found to have a positive effect (B = 0.36, OR = 1.44, *p* < 0.001), indicating that for every one unit increase in elders’ internet trust, the odds ratio of increasing the elderly’s trust in the healthcare system by one additional level increased by 44%.

For the final model, we included the perceptions and financial protection variables, and the results supported these variables as strong significant predictors of the dependent variable. More specifically, for every one unit increase in the elderly’s perceived fairness of the healthcare system, the odds ratio of increasing the elderly’s trust in the healthcare system by one additional level increased by 197% (B = 1.09, OR = 2.97, *p* < 0.001); for every one unit increase in elders’ perception of “Kangbingnan, Kangbinggui”, the odds ratio of increasing elders’ trust by one additional level decreased by 32%. The results indicated that the elderly who perceived fairness were more likely to trust the healthcare system, whereas the elderly who perceived “Kangbingnan, Kangbinggui” were less likely to trust the healthcare system.

## 5. Discussion and Implications

### 5.1. Demographic Variables

Our research explored the Chinese elderly’s trust in the healthcare system and the potential factors affecting their trust in the healthcare system using a cross-sectional population-based survey in China. The findings showed that the Chinese elderly had low trust in the healthcare system. The healthcare system ranked almost at the bottom when respondents evaluated the trustworthiness of 12 institutions, which revealed a trust crisis in the healthcare system among the Chinese elderly.

Our research revealed that most elderly respondents have a low education level, with only 3.1% having received a university specialist degree or above. They also have low annual income. Among the first set of variables, gender and educational level significantly impacted elderly’s trust in the healthcare system. The results indicated that female elders and elders with primary education or below were more likely to trust the healthcare system. A previous study on trust toward physicians in China found that males and people with higher education levels had lower trust toward physicians [23]. Similarly, some empirical studies in other areas also showed that females and those with lower education levels tended to have blind trust in physicians [51]. A study based on the International Social Survey Program found that higher education levels predicted lower trust in the healthcare systems [22]. Our research is in accordance with these studies in terms of disparities in trust associated with gender and educational differences. The possible explanation may be that the female elderly play a traditional gender role, which is more obedient to authorities and professionals. Additionally, those with higher education levels may have more requirements for healthcare providers and may be more critical of public issues.

### 5.2. SES, Social Mobility of SES, and Social Mobility Expectation of SES

Our research focused on subjective SES, perceived mobility of SES, and mobility expectation of SES and these are represented by the self-rated current SES. Besides subjective SES, we also took perceived social mobility of SES and perceived social mobility expectation of SES as the independent variables because social mobility could be used to assess the fairness of the society and is closely related to health [50]. Though the SES variables did not significantly predict trust in the regression models, subjective SES and SES upward mobility expectation were significantly related to trust in the healthcare systems. Those who have higher subjective SES and higher perceived upward social mobility expectation exhibited more trust in the healthcare system. The existing empirical studies in China revealed that subjective SES was highly related to health. Invidious upward social comparisons would lead to perceived relative deprivation, which is negatively associated with self-rated health in both urban and rural older people [52]. Current studies based on a large-scale survey, such as China Family Panel Studies, also showed that subjective social status was correlated with health, personal relative deprivation had negative effects on the health of the Chinese population, and individuals with high expected mobility were found to have significantly better health [50]. Our research further verified that low subjective SES and low mobility expectation of SES led to low trust in the healthcare system. It is possible that the sense of perceived relative deprivation would bring about discontent and dissatisfaction with healthcare services.

### 5.3. Internet Access and Trust in the Internet

Our research is in accordance with CNNIC’s reports that most elderly in China do not have access to the internet. According to the 45th Statistical Report on Internet Development in China released by China Internet Network Information Center (CNNIC) in 2020, until March 2020, the proportion of internet users aged 50 and above was 16.9% [53]. In our research, 72.6% of elderly respondents had no access to the internet. It is not surprising that the elderly without internet access exhibited fair trust in the internet because they had no exposure to misinformation and fake information online. It also seems plausible that the respondents that had higher trust in internet also had higher trust in the healthcare system because these respondents might have a higher propensity for trust. As existing studies show, individuals with a higher propensity for trust will act in a cooperative, prosocial, and moral manner, which can drive the formation of trust [9].

### 5.4. Perceptions of the Healthcare System and Financial Protection

The fourth set of variables is the perceptions of the healthcare system and financial protection. Our research results showed that China has established universal social insurance and pension coverage because most respondents have both old-age and medical care insurance. It seems reasonable that the elderly participating in insurance and pension schemes perceive healthcare as a relatively fair aspect in social life. Although most elders perceived healthcare as fair, our research resonates with the previous studies on the social problem of “Kangbingnan, Kanbinggui” in that the accessibility and affordability of the healthcare system have remained problematic among the Chinese elderly. Over half of the elderly respondents reported healthcare services as difficult to access and unaffordable. It is not contradictory that the elderly respondents perceived fairness in the healthcare system and difficulties in accessibility and affordability simultaneously. According to existing research, the cream-skimming effect in China’s healthcare service is the inequity between the senior cadres and the general public in terms of hospital access and payment [54]. The result implied that the general public encountered similar difficulties in accessibility and affordability.

The results showed that the financial protection of the general public has no great disparities, which might be helpful formulate the elderly’s fairness perception of the healthcare system. However, it should be noted that existing studies found that segmentation by urban–rural and employment status involved different benefits packages provided by China’s healthcare insurance schemes [55] and a stratified pension system for the elderly had also led to inequality in SES [56]. Although our findings show elderly respondents in the CSS database have no great disparities in enrollment in insurance and pension schemes, we call for a more nuanced survey on the actual reimbursement rate of medical expenses and healthcare utilization of the elderly at the individual level in both rural and urban areas, which may influence their financial protection status.

Healthcare accessibility and affordability have been key sources of social discontent in China [57]. Our research found that the perception of “Kanbingnan, Kanbinggui” was significantly associated with the elderly’s trust in the healthcare system. “Kanbingnan, Kanbinggui” has been an unsolved social problem since the end of the 1990s and has driven healthcare reform. Though after decades of endeavors in healthcare reform, universal health insurance may alleviate the perceived unfairness of healthcare supply and distribution [55], and the research findings showed that the Chinese elderly still encountered difficulties with the accessibility and affordability of healthcare services. For those who showed less trust in the healthcare system, the accessibility and affordability of healthcare were still perceived as a social problem, which reveals that there is much to do to improve the supply and distribution of healthcare resources and reduce the medical expenditure of the elderly.

## 6. Conclusions

Our research found that elders’ trust in healthcare was significantly associated with gender, educational level, and internet access. Our research also revealed that the elderly’s trust in the healthcare system was significantly related to their subjective perception of their SES, upward mobility, and perception of accessibility and affordability rather than objective indicators of SES such as income and financial protection. Our research resonated with other studies based on the CSS data which showed that self-rated social economic status, affordability, and perceived quality played significant roles in determining public perceptions of the healthcare system [7]. The longitudinal studies based on the Chinese General Social Survey (CGSS) and CSS have already found that there is a downward socio-psychological tendency for Chinese people to position themselves in the lower strata rather than the middle strata [58,59].

For the elderly, the situation may be more complex. With the decline in both financial and health conditions in later years, the elderly may generate a pessimistic expectation of their subjective social status and future possibilities of upward mobility. They may further form a sense of relative deprivation, which deepens their distrust of the health system. The sense of relative deprivation is generated from individuals’ comparison of themselves with their reference group. The existing studies show that, in China, doctors appear to enjoy lifestyles that their official salaries cannot support, while the internet is awash with expressed sentiments of relative deprivation stemming from the gap between what people receive and what they think they deserve. The relative deprivation has already threatened political trust [60]. The lower the income of the elderly, the more difficult it is to meet their own needs, and they are more likely to be in a disadvantaged position when making social comparisons with others, resulting in relative deprivation [52]. The results of our research implied that lower self-rated SES and upward mobility of the elderly might signal a sense of relative deprivation, which contributed to the low trust in the health system. The inequality in allocation of medical resources, which manifests as cream-skimming effects [54], may also arouse such a sense of relative deprivation.

Our research reveals the role of subjective–perceptive determinants in constructing public trust in the healthcare system. Thus, besides improving the income and financial protection of the elderly through pension and old-age insurance schemes, it is also important to alter the elderly’s subjective perception of inequality in the income gap, distribution of social welfare, and medical resources. Our research findings suggest that there is still much work to do to increase the supply of healthcare services and lower medical expenditure. Assuaging the discontent aroused by “Kanbingnan, Kanbinggui” may effectively increase trust in the healthcare system. At the same time, more efforts should be put into establishing a more balanced report on medical professionals and hospitals to avoid the partial or negative image of the healthcare system amplified by the internet and other media. It is also important to improve the compliance of well-educated male elderly with medical professionals.

The limit of our research is that the impact of regional and urban–rural disparities on SES, financial protection, and accessibility and affordability of healthcare among the elderly was not fully explored. Existing studies revealed that the urban–rural gap and regional gap remain in pension and insurance schemes. The benefits of different healthcare insurance schemes have disparities in urban and rural regions [61]. Many factors, such as Hukou (urban or rural household registration), residential areas, types of healthcare insurance schemes, inpatient and outpatient care, and healthcare utilization at different levels of hospitals (municipalities, counties, or communities), will influence the reimbursement rate of medical expenditure [55]. Males and urban residents who hold an urban Hukou benefit more from China’s current public pension scheme than females and urban residents who hold a rural Hukou [62]. Despite the outstanding achievement of coverage extension, pension inequality has widened among provinces and between pension types. Those who live in advanced areas received more and higher pensions than their counterparts in less developed areas [56]. The launch of a hierarchical medical system in China also brings complexity to the reimbursement gap of healthcare seeking among hospitals at different levels and in different regions [63]. The medical resources are unevenly distributed between urban and rural, as well as among different levels of healthcare providers, which may cause distrust towards the rural primary care system and drive patients to seek treatment in big hospitals in cities [64]. The preference for higher-tiered healthcare providers and generosity in health insurance schemes such as Urban Employee Medical Insurance tend to induce higher health consumption and exacerbate cost escalation for healthcare systems [65]. Thus, the interplay among trust, reimbursement rate, distribution of healthcare resources, and healthcare-seeking behaviors related to the elders will be a potential research topic in the future.

Moreover, during the COVID-19 pandemic, the doctor–patient relationship and trust between patients and medical workers in China increased [66]. The fluctuation of trust in healthcare systems and in medical professionals during the COVID-19 pandemic also needs further studies. Finally, though this study has revealed a representative understanding of internet use among the elderly, this study was not able to reveal an in-depth understanding of the different internet usage types and levels of engagement, and the mechanisms behind such differences. Future research is encouraged to conduct a qualitative study (such as a case study) to unpack the complex phenomenon.

## Figures and Tables

**Table 1 ijerph-19-16461-t001:** Descriptive statistics of the variables of interest.

	Variables	Categories (Value)	Percentages (%)	Mean (SD)
**Dependent variable**				
	Trust in the healthcare system	Trust (1)	68.2	-
		Not trust (0)	25.7	
**Demographics**				
	Gender	Male (1)	47.5	-
		Female (0)	52.5	
	Age	-	-	64.59 (2.74)
	Education level	Primary education or below (1)	54.4	-
		Junior high school (2)	26.8	
		High school (3)	15.3	
		University specialist or above (4)	3.1	
	Total annual income	-	-	18,549.07 (33,065.88)
**SES**				
	SES mobility	-	-	2.08 (0.55)
	Current status	-	-	1.61 (0.65)
	SES mobility expectation	-	-	1.86 (0.78)
**Internet use**				
	Internet access	Yes (1)	27.4	-
		No (0)	72.6	
	Trust in internet	-	-	2.77 (0.60)
**Perception and financial protection**				
	Perceived fairness	-	-	2.84 (0.86)
	Old-age insurance	Yes (1)	78.8	-
		No (0)	21.1	
	Medical insurance	Yes (1)	86.5	-
		No (0)	13.3	
	Kanbingnan, Kanbinggui	Yes (1)	59.0	-
		No (0)	41.0	

Note. “Kanbingnan, Kanbinggui” means “It is difficult to get access to medical services and the cost of medical services is unaffordable”.

**Table 2 ijerph-19-16461-t002:** Descriptive statistics of the elderly’s trust in the healthcare system on the four sets of independent variables.

Variables	Categories (Value)	Trust (N = 1556) N (%)/M (SD)	Distrust (N = 587) N (%)/M (SD)	χ^2^/t
**Demographics**				
Gender	Male	689 (44.3%)	342 (58.3%)	χ^2^ = 33.38 ***
	Female	867 (55.7%)	245 (41.7%)	
Age		64.61 (2.75)	64.46 (2.68)	t = 1.17
Education level	Primary education or below	904 (58.3%)	244 (41.8%)	χ^2^ = 42.97 ***
	Junior high school	383 (24.7%)	202 (34.6%)	
	High school	218 (14.1%)	117 (20.0%)	
	University specialist or above	45 (2.9%)	21 (3.6%)	
**SES**				
Upward mobility	-	2.09 (0.56)	2.08 (0.54)	t = 0.61
Current status	-	1.64 (0.64)	1.55 (0.66)	t = 2.89 *
Upward mobility expectation		1.89 (0.78)	1.78 (0.79)	t = 2.69 *
**Internet use**				
Internet access	Yes (1)	362 (23.3%)	233 (39.7%)	χ^2^ = 57.36 ***
	No (0)	1194 (76.7%)	354 (60.3%)	
Trust in internet		2.81 (0.59)	2.68 (0.62)	t = 4.15 ***
**Perception and financial protection**				
Perceived fairness		3.05 (0.77)	2.29 (0.85)	t = 18.01 ***
Old-age insurance	Yes (1)	1225 (78.8%)	472 (80.5%)	χ^2^ = 0.76
	No (0)	329 (21.2%)	114 (19.5%)	
Medical insurance	Yes (1)	1347 (86.8%)	506 (86.2%)	χ^2^ = 0.13
	No (0)	205 (13.2%)	81 (13.8%)	
Kanbingnan, Kanbinggui	Yes (1)	859 (55.1%)	402 (68.5%)	χ^2^ = 31.32 ***
	No (0)	698 (44.9%)	185 (31.5%)	

Note. * *p* < 0.05, *** *p* < 0.001; χ^2^ represents chi-square; “Kanbingnan, Kanbinggui” means “It is difficult to get access to medical services and the cost of medical services is unaffordable”.

**Table 3 ijerph-19-16461-t003:** Binary logistic regression models for the elderly’s trust in the healthcare system.

	Model 1			Model 2			Model 3			Model 4		
	B (OR)	SE	95%CI	B (OR)	SE	95%CI	B (OR)	SE	95%CI	B (OR)	SE	95%CI
**Demographics**												
Gender	−0.47 *** (0.63)	0.11	0.50–0.79	−0.46 ** (0.63)	0.11	0.50–0.79	−0.49 *** (0.61)	0.13	0.48–0.78	−0.53 *** (0.59)	0.14	0.45–0.78
Age	0.002 (1.00)	0.02	0.96–1.04	−0.004 (1.00)	0.02	0.96–1.04	−0.01 (0.99)	0.02	0.95–1.03	0.01 (1.01)	0.03	0.96–1.06
Education level												
Junior high school	−0.24 (0.79)	0.17	0.56–1.11	−0.22 (0.65)	0.17	0.57–1.13	−0.09 (0.92)	0.20	0.62–1.37	−0.05 (0.95)	0.23	0.60–1.52
High school	−0.64 *** (0.53)	0.17	0.38–0.73	−0.63 *** (0.53)	0.17	0.38–0.74	−0.11 (0.73)	0.20	0.50–1.09	−0.33 (0.72)	0.23	0.46–1.14
University specialist or above	−0.40 (0.67)	0.32	0.36–1.26	−0.43 (0.65)	0.33	0.35–1.24	0.10 (1.11)	0.35	0.55–2.21	0.29 (1.34)	0.39	0.62–2.89
Total annual income	<0.01 (1.00)	<0.01	1.00–1.00	<0.01 (1.00)	<0.01	1.00–1.00	<0.01 (1.00)	<0.01	1.00–1.00	<0.01 (1.00)	<0.01	1.00–1.00
**SES**												
Upward mobility				0.13 (1.13)	0.10	0.93–1.38	0.21 (1.24)	0.11	1.00–1.53	0.13 (1.14)	0.12	0.90–1.44
Current SES				0.04 (1.04)	0.11	0.83–1.30	−0.01 (0.99)	0.12	0.79–1.27	−0.07 (0.93)	0.14	0.71–1.22
Upward mobility expectation				0.12 (1.13)	0.09	0.96–1.33	0.13 (1.14)	0.09	0.95–1.37	0.07 (1.07)	0.10	0.87–1.31
**Internet use**												
Internet access							−0.58 *** (0.56)	0.13	0.43–0.73	−0.47 *** (0.62)	0.15	0.47–0.83
Trust in internet							0.36 *** (1.44)	0.10	1.18–1.75	0.245 *** (1.28)	0.11	1.02–1.60
**Perception and financial protection**												
Perceived fairness										1.09 *** (2.97)	0.09	2.51–3.52
Old-age insurance										0.05 (1.05)	0.05	0.74–1.50
Medical insurance										0.08 (1.08)	0.08	0.72–1.63
Kanbingnan, Kanbinggui										−0.38 *** (0.68)	0.14	0.52–0.89
N	2134			1785			1490			1414		
Nagelkerke R^2^	0.05			0.05			0.08			0.27		

Note. ** *p* < 0.01; *** *p* < 0.001; “Kanbingnan, Kanbinggui” means “It is difficult to get access to medical services and the cost of medical services is unaffordable”.

## Data Availability

This study was authorized to use a publicly archived dataset from the Chinese Social Survey 2019 (link: http://csqr.cass.cn/index.jsp).

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
