# Peer review of "Exploring Chinese Elderly’s Trust in the Healthcare System: Empirical Evidence from a Population-Based Survey in China"

_ijerph, 2022, doi:10.3390/ijerph192416461_

Round 1

Reviewer 1 Report

under the methods section, the authors mention "institutions" quite a number of time, but failed to explain what these institutions are. This might make it difficult for readers to understand the study. for isntance, you mention 12 different institutions, please list them and explain their relevance to this study.

Also, since internet usage is a major variable in yourr study, it would be beenficial to exlpain somewhere in the literature the level of internet usage among the elderly population. 

Author Response

Under the methods section, the authors mention "institutions" quite a number of time, but failed to explain what these institutions are. This might make it difficult for readers to understand the study. for instance, you mention 12 different institutions, please list them and explain their relevance to this study.

Thanks for your comment. The institutions refer to the 12 institutions listed in the CSS 2019 survey, and the healthcare system is listed among them. We have this information added in lines 196 to 201 as follows:

“These 12 institutions not only cover the official apparatus, public sectors, private sectors, but also contain specific workplaces of the respondents. Besides the healthcare system such as hospital, the other institutions include the “central government; government at district or county-level; government at town-level; organizations such as labor union, the Communist Youth League and the women’s federation; your working unit or company; charity organizations; media; bank; insurance company; court; police.”

Also, since internet usage is a major variable in your study, it would be benficial to explain somewhere in the literature the level of internet usage among the elderly population.

Thanks for your comment. In our research, we examine the “Internet use” variables, and we also did not adopt the term “Internet usage”. Although there are many studies related to Internet usage among the elderly (for example, the studies on digital divide and digital exclusion, etc.), this study focused on Internet use among the elderly. In line 244 to 245, we further explained that Internet use variables include access to the Internet and trust in the Internet. More specifically, Internet use refers to the elderly’s access to the Internet or not in their daily life, with two response categories as “Yes” or “No”; Trust in the Internet refers to the elderly’s trust perception in the Internet, with different levels of trust were captured by a 4-point Liker scale. In another word, whereas Internet access refers to the elderly’s physical access to the Internet, Internet trust reflects the elderly’s perceptions of Internet use.

We agree with the reviewers that it will be interesting to investigate further the usage levels of the elderly. Nevertheless, this study was based on the open-source CSS 2019 survey, and thus it is beyond the scope of this study to add more questions to explore the different usage levels of the Internet among the elderly or to conduct a further qualitative study on the elderly to reveal how and why they have different usage levels of the Internet. We have added this as the research limitations and a future study in lines 550 to 554 in the Conclusion section.

“Finally, though this study has revealed a representative understanding of Internet use among the elderly, this study was not able to reveal an in-depth understanding of the different Internet usage types and levels of engagement, and the mechanisms behind such differences. Future research is encouraged to conduct a qualitative study (such as a case study) to unpack the complex phenomenon.”

Reviewer 2 Report

The manuscript entitled “Exploring Chinese elderly’s trust in the healthcare system:
Empirical Evidence from a Population-based Survey in China” is well written and properly explained. The article is well structured with comprehensive approach and easy to understand. Authors have introduced the issue in focus in detail with good references. Statistical analysis of the data is well explained and covers almost all aspects of the issue in hand. Results are described in a manner, that are easy to comprehend and interpret. In conclusion the authors have given valuable suggestions about factors, which should be kept in mind regarding the improvement of trust of elderly in healthcare system. However, in my opinion COVID-19 pandemic scenario should also be discussed in preview of a factor that can also effect the trust of population on healthcare system of a country. Best wishes and regards, 

Author Response

Reviewer 2

The manuscript entitled “Exploring Chinese elderly’s trust in the healthcare system:

Empirical Evidence from a Population-based Survey in China” is well written and properly explained. The article is well structured with comprehensive approach and easy to understand. Authors have introduced the issue in focus in detail with good references. Statistical analysis of the data is well explained and covers almost all aspects of the issue in hand. Results are described in a manner, that are easy to comprehend and interpret. In conclusion the authors have given valuable suggestions about factors, which should be kept in mind regarding the improvement of trust of elderly in healthcare system. However, in my opinion COVID-19 pandemic scenario should also be discussed in preview of a factor that can also effect the trust of population on healthcare system of a country. Best wishes and regards,

Thanks for your comment. We also think the COVID-19 pandemic has a profound influence on trust in the healthcare system and the doctor-patient relationship in China. However, the database of 2019 CSS is collected in 2019, and the COVID-19 pandemic began in 2020. From the 2019 CSS database, we cannot analyze the influence of the COVID-19 pandemic on trust in the healthcare system. To address the question, we add an existing study on changing the doctor-patient relationship during the COVID-19 pandemic as a direction for future studies in the Conclusion section.

We added lines 547 to 550 as follows:

“Moreover, during the COVID-19 pandemic, the doctor-patient relationship and trust between patients and medical workers in China increased [66]. The fluctuation of trust in healthcare systems and in medical professionals during the COVID-19 pandemic also needs further studies.”

Reviewer 3 Report

In the present manuscript, the authors studied how the Chinese elderly trust the healthcare system and found that gender, educational level, and Internet access were the critical factors that significantly affect elders’ trust in healthcare system. The research is important in the field particularly for the existence of trust crisis for healthcare system among the public. However, there are few minor concerns that should be addressed:

1.    Line 16-Expand SES.

2.    Line 126-Add relative reference

3.    Table 1 and Line 296-Don’t understand what the meaning is “Kanbingnan, Kanbinggui”. The authors should use suitable description

4.    Table 2-There is no”**” in Table body.

Table 3-There is no”*” in Table body.

Table 3 data need to be arranged properly.

Author Response

Reviewer 3

In the present manuscript, the authors studied how the Chinese elderly trust the healthcare system and found that gender, educational level, and Internet access were the critical factors that significantly affect elders’ trust in healthcare system. The research is important in the field particularly for the existence of trust crisis for healthcare system among the public. However, there are few minor concerns that should be addressed:

  1.    Line 16-Expand SES.
  2.    Line 126-Add relative reference
  3.    Table 1 and Line 296-Don’t understand what the meaning is “Kanbingnan, Kanbinggui”. The authors should use suitable description
  4.    Table 2-There is no”**” in Table body.

Table 3-There is no”*” in Table body.

Table 3 data need to be arranged properly.

  1. Thanks for your notice. We have expanded SES to social-economic status in the abstract (line 16).
  2. The statement in line 126 is a topic sentence of the paragraph, which sums up the main point of the paragraph. Following the topic sentence, we have presented an elaborated discussion of the main idea with relevant references. We agree that the statement should be sharpened to be more outstanding as a topic sentence, and we have improved the statement (from lines 125 to 127 now) as follows:

“As the proliferation of the Internet has dramatically changed Chinese society, the influence of exposure to online information on trust in the healthcare system became an area worth studying.”

  1. We have described the term “Kanbingnan, Kanbinggui” in lines 156 to 163 as follows:

“In China, the discontent from the public towards accessibility and affordability of healthcare services is commonly known as “Kanbingnan, Kanbinggui” in Chinese, which literally means “It is difficult to get access to medical services and the cost of medical services is unaffordable” [44].”

Given that the unique term “Kanbingnan, Kanbinggui”originated from the Chinese cultural context and we could not replace it with a suitable term that can fully reflect its actual meaning after a thorough literature review, we would still keep this description in the manuscript. Also, CSS 2019 questionnaire adopted the same term. Nevertheless, we understand that this may pose difficulties to the readers’ understanding. We have added notes of its meaning under each table in which it was presented. (line 297 to 298; line 328 to 329; line 382 to 383)

  1. Thanks for your notice. We have removed “**P<.01” from the note in Table 2 and Table 3. We have also rearranged the data in Table 3 to a better look.

Reviewer 4 Report

“Exploring Chinese Elderly Trust” is a good manuscript and I recommend it should be published.  It is methodologically sound and it adds to our knowledge about Chinese health care delivery.

There are many aspects to this manuscript that I liked. The literature review is very comprehensive and it does a nice job in terms of distinguishing macro versus micro factors impacting trust in general. It then does a good job in pointing to how China’s population is aging  and therefore looking at elder attitudes to healthcare delivery is an important and salient question. Thus, the literature review nicely sets up the research question in nothing the absence of existing research looking at elder attitudes in China regarding trust.

The selection of demographic variables is good. I also like how the authors use mixed methods such as T-Tests and regression to look for significance for which variables are significant.

Finally the discussion and presentation of results are sound. Overall a good manuscript.

I do have suggestions.  First, I would be curious to see if geography (urban and rural) are salient variables.  Two, the author could do a better job exploring the significance of their findings. By that, given the increasing age of the Chinese population and also possible economic growth slow downs that are occurring, what might all this mean for attitudes toward Chinese health care delivery in the future. Do these changes and the results of this study suggest policy options, reforms, or crises.  Finally, how do the results here contribute to our knowledge about healthcare in China or theories about trust not just in China but perhaps elsewhere in the world.  Is it possible the results here can be generalized to other countries?

Author Response

Reviewer 4

“Exploring Chinese Elderly Trust” is a good manuscript and I recommend it should be published.  It is methodologically sound and it adds to our knowledge about Chinese health care delivery.

There are many aspects to this manuscript that I liked. The literature review is very comprehensive and it does a nice job in terms of distinguishing macro versus micro factors impacting trust in general. It then does a good job in pointing to how China’s population is aging and therefore looking at elder attitudes to healthcare delivery is an important and salient question. Thus, the literature review nicely sets up the research question in nothing the absence of existing research looking at elder attitudes in China regarding trust.

The selection of demographic variables is good. I also like how the authors use mixed methods such as T-Tests and regression to look for significance for which variables are significant.

Finally the discussion and presentation of results are sound. Overall a good manuscript.

I do have suggestions. First, I would be curious to see if geography (urban and rural) are salient variables. Two, the author could do a better job exploring the significance of their findings. By that, given the increasing age of the Chinese population and also possible economic growth slow downs that are occurring, what might all this mean for attitudes toward Chinese health care delivery in the future. Do these changes and the results of this study suggest policy options, reforms, or crises. Finally, how do the results here contribute to our knowledge about healthcare in China or theories about trust not just in China but perhaps elsewhere in the world.  Is it possible the results here can be generalized to other countries?

Thank you for your inspiring suggestions. We did not include urban and rural disparities in our variables because there have already been many studies on this topic, and moreover, we think the complexity in urban and rural disparities cannot be analyzed according to 2019 CSS database. In line 463 to 465, we quote some of existing studies. Then we add a statement in line 468 to 472 to point out what kind of information we need to explore urban-rural disparities as follows:

 “Although our findings show elderly respondents in the CSS data base have no great disparities in enrollment in insurance and pension schemes, we call for a more nuanced survey on the actual reimbursement rate of medical expenses and healthcare utilization of the elderly at the individual level in both rural and urban areas, which may influence their financial protection status.”

We have to admit that 2019 CSS database questionnaire contains no information about the actual reimbursement rate, health status and healthcare utilization at individual level, then it is difficult for us to make conclusion about the impacts of urban or rural insurance schemes on people’s perception or trust in healthcare system (For example, according to current policy, if a person seeks healthcare in primary care center in his or her community or village, then most of medical cost will be covered no matter urban or rural insurance schemes he or she enrolled in. However, if he or she seek healthcare at a famous hospital outside his or her province, then the reimbursement rate will be quite low no matter what kind of insurance schemes he or she enrolled in. Thus, different healthcare seeking behaviors, different insurance packages for urban and rural residents contribute to different reimbursement rate and then may cause different financial burden. With these details, then it is possible for us to discuss the regional disparities or urban-rural dualities and their mediating role in trust. However, in CSS 2019 database, the questionnaire did not contain such information. Existing studies based on China Household Finance Survey (CHFS) and China Health and Retirement Longitudinal Survey (CHARLS) have more detailed discussion on urban-rural disparities, but the latter stopped updating in recent years. We have quoted these existing studies as references as limits of our research at the last part of our conclusion. The passage we added at last of our conclusion in line 523 to 546 as follows:

“The limit of our research is not fully exploring the impact of regional and urban-rural disparities on SES, financial protection and accessibility and affordability of healthcare among the elderly. Existing studies revealed that the urban-rural gap and regional gap remains in pension and insurance schemes. The benefits of different healthcare insurance schemes have disparities in urban and rural regions [61]. Many factors, such as Hukou (urban or rural household registration), residential areas, types of healthcare insurance schemes, inpatient and outpatient care, and healthcare utilization at different levels of hospitals (municipalities, counties, or communities) will influence the reimbursement rate of medical expenditure [55]. Males and urban residents who hold an urban Hukou benefit more from China’s current public pension scheme than females and urban residents who hold a rural Hukou [62]. Despite the outstanding achievement of coverage extension, pension inequality has widened among provinces and between pension types. Those who live in advanced areas received more and higher pensions than their counterparts in less developed areas [56]. The launch of a hierarchical medical system in China also brings complexity in the reimbursement gap of healthcare seeking among hospitals at different levels and in different regions [63]. The medical resources are unevenly distributed between urban and rural, as well as among different levels of healthcare providers, which may cause distrust towards the rural primary care system and drive patients to seek for treatment in big hospitals in cities [64]. The preference of higher-tiered healthcare providers and generosity in health insurance schemes such as Urban Employee Medical Insurance tend to induce higher health consumption and exacerbate cost escalation for health care systems [65]. Thus, the interplay among trust, reimbursement rate, distribution of healthcare resources and healthcare-seeking behaviors related to the elders will be a potential research topic in the future.”   

Our results may not be able to be generalized to other countries. First, different countries have different retirement, pension, medical insurance scheme and healthcare system. Second, China is undergoing a rapid transition in population, family structure and medical reform. Nevertheless, we did find some similarities between our results and existing studies based on an international survey (lines 403 to 406). Also, we clarify the contribution of our research to the knowledge of trust by adding the following sentence in lines 511 to 512 “Our research reveals the role of subjective-perceptive determinants in constructing public trust towards the healthcare system”.

We also changed the reference number in line 403 from [22] to [23] after checking.